# Fetal Brain Tumors, a Challenge in Prenatal Diagnosis, Counselling, and Therapy

**DOI:** 10.3390/jcm12010058

**Published:** 2022-12-21

**Authors:** Ivonne Alexandra Bedei, Thierry A. G. M. Huisman, William Whitehead, Roland Axt-Fliedner, Michael Belfort, Magdalena Sanz Cortes

**Affiliations:** 1Department of Prenatal Diagnosis and Fetal Therapy, Justus-Liebig University Giessen, 35392 Giessen, Germany; 2Division of Fetal Therapy and Surgery, Department of Obstetrics and Gynecology, Texas Children’s Hospital Fetal Center, Baylor College of Medicine, Houston, TX 77030, USA; 3Edward B. Singleton Department of Radiology, Texas Children’s Hospital, Baylor College of Medicine, Houston, TX 77030, USA; 4Department of Neurosurgery, Baylor College of Medicine, Houston, TX 77030, USA; 5Department of Obstetrics and Gynecology, Texas Children’s Hospital Pavilion for Women, Baylor College of Medicine, Houston, TX 77030, USA

**Keywords:** fetal brain tumor, macrocephaly, teratoma, hydrocephalus, prenatal imaging

## Abstract

Fetal brain tumors are a rare entity with an overall guarded prognosis. About 10% of congenital brain tumors are diagnosed during fetal life. They differ from the postnatally encountered pediatric brain tumors with respect to location and tumor type. Fetal brain tumors can be benign or malignant and infiltrate or displace adjacent brain structures. Due to their high mitotic rate, they can show rapid growth. Outcome depends on age of diagnosis, size, and histological tumor type. Findings like polyhydramnios and macrocephaly encountered on routine ultrasound are frequently associated. Detailed prenatal anomaly scan and subsequent fetal magnetic resonance imaging (MRI) may identify the brain tumor and its severity. Both maternal and fetal prognosis should be included in prenatal counselling and decision making.

## 1. Introduction

Fetal brain tumors (FBT) are very rare. They occur in 0.34/1,000,000 live births and account for 10% of all antenatal tumors [1,2,3,4,5]. About 10% of congenital brain tumors are diagnosed during fetal life [4,6]. In comparison to tumors appearing in the pediatric population, they differ in location, histological type, clinical behavior and prognosis [1,4,5,7,8]. Whereas pediatric tumors are mostly infratentorial, FBTs originate typically in the supratentorial region [1,3,9]. Teratomas are the most frequently encountered intracranial tumors in the prenatal period, occurring in about half of the reported cases, in contrast to pilocytic astrocytomas, malignant gliomas and medulloblastoma which are typically found in the pediatric age group [1,3,9,10,11,12].

Depending on the tumor type, diagnosis is usually made in the late second or third trimester [2,3,10]. Intraparenchymal tumors like teratomas, gliomas and embryonal tumors are often characterized by their rapid growth and infiltration of the surrounding brain tissue [7]. Fetal brain tumors most often occur in isolation but can be part of a genetic/tumor predisposition syndrome like Gorlin syndrome (medulloblastoma), Li-Fraumeni syndrome (glioblastoma), Pallister Hall syndrome (hypothalamic hamartoma), neurofibromatosis (astrocytoma) or Rhabdoid predisposition syndrome (ATRT, medulloblastoma, choroid plexus tumors and former PNET) [7,13]. Prognosis is generally poor with a postnatal survival rate varying between 16% and 28% [1,2,9]. Outcome depends on the tumor histological type, growth behavior and the timing of occurrence/diagnosis relative to the gestational age [9]. Limited by the fact that definitive histopathological confirmation is only made after birth, prenatal counselling is difficult and relies on prenatal imaging, combining detailed ultrasound (US) and complementary Magnetic Resonance Imaging (MRI) as well as multiple biometric and clinical parameters. MRI helps to confirm and refine diagnosis and adds valuable predictive information about outcome. Counselling should be multidisciplinary and include both, fetal prognosis, and possible maternal complications.

## 2. Case Series

We present four fetuses with matching prenatal US and MRI diagnosis of FBTs presenting at our high complexity tertiary referral center between 2011–2019. Goal of this case series is to familiarize physicians with the various primary brain tumors that can be encountered on prenatal imaging and the respective value of US and MRI to narrow down differential diagnosis.

Case 1: A 35-year-old G1P1 was diagnosed at 18 weeks and 6 days gestational age with a fetal brain mass of unclear etiology at her referring institution. Outside fetal MRI showed a 2.5 × 4.2 × 3.9 cm suprasellar, predominantly solid, partially cystic midline mass lesion elevating the floor of the third ventricle, high grade compression and displacement of the mesencephalon and brainstem with resultant supratentorial ventriculomegaly and macrocephaly (Figure 1A,B).

Follow-up ultrasound at 20 weeks and one day showed a rapid enlargement in the predominantly solid component of the intracranial mass, measuring 5 cm × 6.3 cm × 7 cm, macrocephaly with a head circumference (HC) of 285 mm (>5SD above the mean) and polyhydramnios (amniotic fluid index, AFI: 26 cm) (Figure 2A,B).

Prominent arterial feeders originating from the middle cerebral artery (MCA) were noted. The MCA peak systolic velocity was >2 MoM (multiple of the median) suggesting a high cardiac output status or fetal anemia. Cardiothoracic ratio was normal. There were no signs of fetal hydrops. Based on the imaging characteristics a germ cell tumor, e.g., teratoma was considered the most likely diagnosis. The parents received extensive multidisciplinary counselling on fetal prognosis and potential maternal complications, mostly due to the large head size. They opted for palliative/comfort care at delivery. Pregnancy was complicated by a preterm premature rupture of membranes (PPROM) with subsequent delivery at 25 weeks outside of our institution. The newborn died at 1 h of life. Histopathological results and mode of delivery are unfortunately unavailable.

Case 2: A 17-year-old G1P0 was referred at 35 weeks and 5 days of gestation to our institution. Ultrasound showed a highly perfused mass lesion located in the posterior fossa associated with hydrocephalus, macrocephaly (HC 366.3 mm, >5 SD above the mean) and mildly increased amniotic fluid volume (AFI 23 cm). On fetal MRI, a well perfused posterior fossa infratentorial mixed solid and cystic mass lesion was seen associated with moderate to marked supratentorial hydrocephalus. Cystic elements of the lesion were seen herniating through the tentorial incisura into the supratentorial compartment. Diffusion weighted imaging (DWI) showed restricted diffusion within the solid tumor component, which could be indicative of a medulloblastoma versus an atypical terato-rhabdoid tumor (ATRT) (Figure 3A–D).

Patient delivered two weeks later (38 weeks) by cesarean section after PROM. A male newborn, with a weight of 3090 g and Apgar scores of 5 and 6 at respectively 1 and 5 min of life was admitted to our neonatal intensive care unit (NICU). MRI performed on the first day of life showed a 50% increase in tumor size, measuring 7 cm in maximal diameter, compressing, and infiltrating nearly the complete cerebellum and vermis as well as much of the midbrain structures. High grade obstructive supratentorial ventriculomegaly with a cystic intraventricular tumor component was again observed. The cystic component appeared T1-hyperintense indicating the high proteinaceous content. The solid tumor component showed multiple serpiginous flow-related signal voids (Figure 4A–C).

After multiple multidisciplinary discussions, it was determined that the tumor was inoperable, and parents opted for comfort care. Baby died on the second day of life from respiratory insufficiency. Parents declined autopsy.

Case 3: A 35-year-old G3P2 pregnant patient was referred to our institution because of a suspected fetal brain tumor at 38 weeks of gestation after initial suspicion for this diagnosis at 34 weeks. Ultrasound and fetal MRI at referral revealed a well circumscribed T2-hypointense solid cortical/subcortical lesion in the left occipital lobe measuring 4.3 × 2.9 × 3.1 cm (Figure 5A,B), which appeared larger than on the imaging performed four weeks earlier.

The lesion exerted mild local mass effect on the left occipital horn without midline shift or hydrocephalus. On US, the lesion was mildly heterogeneously hyperechogenic and presented with an asymmetrically increased arterial flow in the left common carotid, middle cerebral and posterior cerebral arteries. There was no evidence for associated hemorrhage. Polyhydramnios was not observed. A neuroglial tumor with the differential diagnosis of cortical dysplasia was suspected. A female newborn was delivered via elective C-section. MRI and computer tomography (CT) scan performed on day one of life showed a 4.7 cm × 3 mm × 3.5 cm mass lesion in the left occipital lobe with T2-hypointense susceptibility related signal loss due to areas of microcalcification (Figure 6A–C).

Resection was performed on day 5 of life, followed by chemotherapy. Histopathology demonstrated a glioneural tumor with varying histomorphologies and anaplastic features. The patient successfully completed 12 months of chemotherapy with Cytoxan, Carboplatin, Etoposide and Vincristine following an individualized protocol. She is now 10 years old without tumor recurrence. Neuro-oncology follow-up was discontinued after 7 years without recurrence of disease. She is meeting normal neurodevelopment milestones.

Case 4: A 26 years old G2P1 patient presented at 28 weeks for further diagnostic work-up for ventriculomegaly in twin B in a dichorionic diamniotic twin pregnancy. Ultrasound at 31 weeks and 1 day showed a complex cystic-solid mass lesion in the left frontal lobe measuring 33 × 34 × 38 mm surrounded by echogenic brain parenchyma. Additionally, irregular echogenic borders of the third ventricle and lateral ventricles were observed suggesting intraventricular hemorrhage (Figure 7A). HC was within the normal range (73rd percentile), bilateral ventriculomegaly was noted (left and right atrial width measured 18 mm and 11 mm, respectively) (Figure 7A–C). Fetal MRI showed a heterogenous mass lesion predominately located within the left frontal lobe, containing a solid component with internal vascularity as well as a large hemorrhagic component. (Figure 7D). Image findings were considered most consistent with ATRT. Cesarean delivery was performed at 35.6 weeks because of preterm labor and the transverse position of twin B. Twin B, a female newborn with 2330 gr and Apgar score 8/8 at, respectively 1 and 5 min was delivered and transferred to the NICU on CPAP after delivery. MRI at the same day showed interval enlargement (5.6 cm × 6.5 cm × 4.7cm) of the left frontal lobe cystic-solid mass lesion extending into the suprasellar cisterna and ventricular system (Figure 7E). Multifocal hemorrhagic products with intralesional susceptibility effects were seen, the solid component of the tumor showed restricted diffusion and enhancement. The lesion was inseparable from the optic chiasm, optic nerves, and pituitary stalk. It incased the bilateral supraclinoid internal carotid arteries (ICAs), bilateral anterior cerebral arteries (ACAs), right proximal middle cerebral artery (MCA), distal basilar artery, and its branches. Biopsy and partial tumor resection were performed on day 5 of life, which confirmed an ATRT, WHO grade IV with heterogenic staining of the INI1 marker. Whole exome sequencing (WES) studies was negative for any known deleterious mutation. In the light of the very guarded prognosis and difficulties for radical neurosurgical tumor resection secondary to the encasement of much of the circle of Willis, parents opted for palliative care. The infant developed seizures during home care. To reassess the tumor and rule out progressing hydrocephalus, repeat MRI was done. Despite the expected poor prognosis, follow up MRI revealed a near complete involution of the solid tumor with multiple loculated cerebrospinal fluid (CSF) intensity fluid collections occupying much of the frontal lobe. There were no nodular areas of enhancement suggestive of residual tumor (Figure 7F) except for two non-enhancing angular areas of hemorrhagic material.

Biopsy of the nodules, supratentorial ventriculostomy (ETV) with placement of an externally draining ventricular drain was performed. Pathology did not show viable tumor tissue. Subsequent need for the placement of a permanent ventriculoperitoneal (VP) shunt was indicated. This is a very unusual course, published and discussed by Peterson et al. [14]. The girl is currently 9-year-old. She suffers from global developmental delay, epilepsy, cerebral palsy and is mostly fed via a percutaneous G-tube.

## 3. Discussion

Our case series exemplifies various entities of FBT, the course of pregnancy and their postnatal outcome. Mortality is high and overall survival rate can be as low as 16%, depending on the histopathology, location and growth pattern of the tumor [10]. In our case series, two infants suffered neonatal demise (case 1 + 2). Autopsy and histopathological confirmation of prenatally suspected tumor was rejected in these cases, limiting diagnosis to pre- and postnatal imaging. This is in line with the current literature, showing that only 30–40% of parents agree to autopsy after the loss of their infant and highlights the need for alternative, less invasive options [15,16].

The gold standard for diagnosis of FBTs is histopathological examination. Molecular analysis can aid for further accurate classification [4]. Both can only be done safely after birth [7]. During gestation, diagnosis, counselling and treatment decisions primarily rely on prenatal imaging, clinical features and progression of disease [7]. FBTs are typically first suspected or identified on prenatal US screening. Findings like hydrocephalus, macrocephaly, and polyhydramnios are often the first sign and lead to further detailed imaging. Additional assessment by fetal MRI may improve diagnostic accuracy [2,3,7,10]. Depending on the gestational age, the diagnostic accuracy may increase up to 29% with a matching increase in the diagnostic confidence [3,13,17]. Differential diagnosis of FBT includes a focal intraparenchymal hemorrhage, which can mimic a mass lesion with or without hydrocephalus, or, possibly secondary to a vascular malformation [4,18].

Associated congenital anomalies can be present in 14–20% of cases, including anomalies of the corpus callosum in case of a pericallosal/midline lipoma and cleft lip and palate in case of teratoma [1]. Brain tumors can be part of a tumor predisposition syndrome or genetic disarrangement and genetic testing should be considered, even more in the presence of associated anomalies [4,5,13,18,19,20]. Different tumor types can be found before birth and exhibit a different prognosis for the fetus itself and the mother to be. The most common prenatal tumor types are teratoma, astrocytoma, craniopharyngioma, choroid plexus papilloma and embryonal tumors like medulloblastoma or ATRT [3,6,10,13,21]. However, reported frequencies of tumor types vary significantly in the existing literature [4,10,18,21]. With the advancing role of molecular diagnostics in tumor classification, implemented by the WHO classification of 2016 and updated in 2021, new tumor types are introduced, and former entities deleted. The reported frequencies of various congenital tumor types may consequently change in the future [4,22,23,24].

In our case series, we observed one case with suspected teratoma, one case with confirmed and one with suspected ATRT and one case with a glioneural tumor. Unfortunately, in 2/4 cases we do not have histopathological confirmation of the tumor histology. The differential diagnosis was based on pre- and postnatal MRI (case 1 + 2). Even though, post mortem autopsy is the gold standard, in case of parental decline, high quality prenatal imaging, and also postnatal or postmortem MRI add important information and can help to narrow down differential diagnosis [25,26]. A universal algorithm for follow-up and delivery planning after diagnosis of a FBT does currently not exist and sonographic follow-up may vary between several times a week to every 4 weeks [7].

In our experience, diagnosis of teratomas occurred early in pregnancy at 18 weeks GA (case 1). This is in accordance with the current literature, reporting that tumors found before 22 weeks GA are most often teratomas with an average age at diagnosis of 27 weeks, while gliomas and other entities are usually diagnosed after 32 weeks of pregnancy [7,13]. The primary tumor location or epicenter of the mass lesion may allow to narrow down the differential diagnosis but the invasive nature and frequently observed rapid growth may prevent or limit correct diagnosis.

Macrocephaly and obstructive, tumor related ventriculomegaly was present in 2/4 cases and are frequent concomitant findings [10,19]. They can lead to cephalopelvic disproportion and halt labor progression and delay delivery [27]. Rarely a spontaneous rupture of the fetal head or uterine rupture during delivery can happen [13]. Anomalous fetal position is frequently encountered [10]. Depending on the gestational age, head circumference and tumor characteristics, Cesarean-section is frequently advised [1,3,5,28]. When palliative care is elected at the time of delivery and there is significant hydrocephalus, a cephalocentesis can be considered to allow for a vaginal birth [29].

Maternal mirror syndrome is described in extremely severe cases of high-cardiac output failure leading to fetal hydrops [13,30].

Polyhydramnios is a well described finding in FBT. It occurs in up to one third of all cases and was seen in 2/4 cases of our series [10,18]. One of the etiologies can be related to high cardiac output status, as seen in case one, which also presumably led to PPROM and preterm delivery. Polyhydramnios may also result from impaired fetal swallowing due to brainstem dysfunction.

Prognosis is generally poor and worsens with increasing tumor size and decreasing gestational age at diagnosis [10]. Demise may occur in utero or after delivery. The options for prenatal therapy are very limited and mostly restricted to the treatment of concomitant symptoms, such as drainage of the polyhydramnios or cephalocentesis in case of severe tumor associated hydrocephalus/macrocephaly before birth. Recently, prenatal treatment of subependymal giant cell astrocytomas in fetuses with tuberous sclerosis with mTOR inhibitors has been reported [31]. Postnatal therapeutic options vary between palliation/comfort care and complete tumor resection with or without shunt placement and chemotherapy with a curative intent [27]. 

## 4. Conclusions

FBTs are a very rare entity. Prenatal counselling using high quality prenatal ultrasound, fetal MRI and clinical features should be multidisciplinary. In the light of the generally guarded fetal prognosis, associated increased risk of maternal morbidity and mortality during ongoing pregnancy and delivery, early termination of pregnancy may be discussed. However, as exemplified by case four, fetal prognosis, tumor development and life expectancy may be different than previously thought and readjustment of treatment plan may be necessary after birth. Histopathological confirmation of the prenatally suspected diagnosis is the gold standard and molecular analysis is increasingly used to further classify the tumor. Because of the rarity of FBTs, most conclusions and recommendations are based on case series. Larger multicenter studies are needed to provide better information on prenatal treatment options, pregnancy monitoring, and postnatal outcome in terms of survival and neurodevelopment.

## Figures and Tables

**Figure 1 jcm-12-00058-f001:**
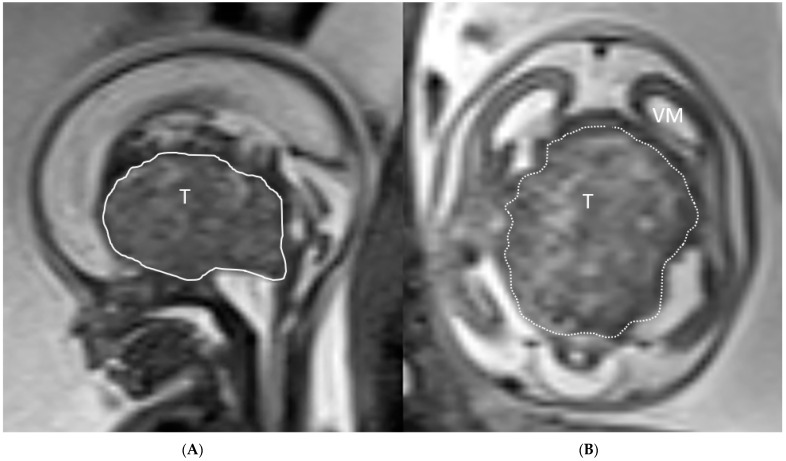
(**A**,**B**): Ultrafast T2-weighted fetal MR image showing a 3.9 cm suprasellar mixed solid-cystic, suprasellar mass lesion compressing the brainstem (BS), mesencephalon (M) and elevating and compressing the third ventricle with resultant supratentorial ventriculomegaly (VM). Imaging characteristics most compatible with a teratoma (T).

**Figure 2 jcm-12-00058-f002:**
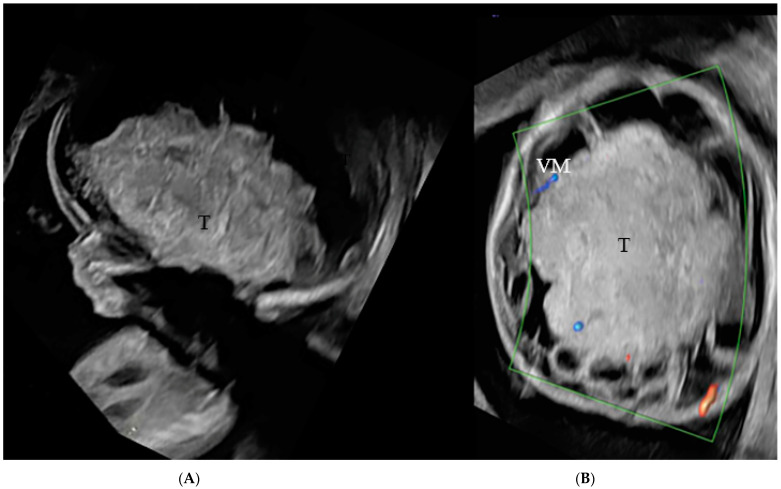
(**A**,**B**): US showing a 5 × 6.3 × 7 cm predominantly hyperechogenic solid midline mass lesion, centered in the suprasellar region (T). High grade supratentorial ventriculomegaly (VM) is noted.

**Figure 3 jcm-12-00058-f003:**
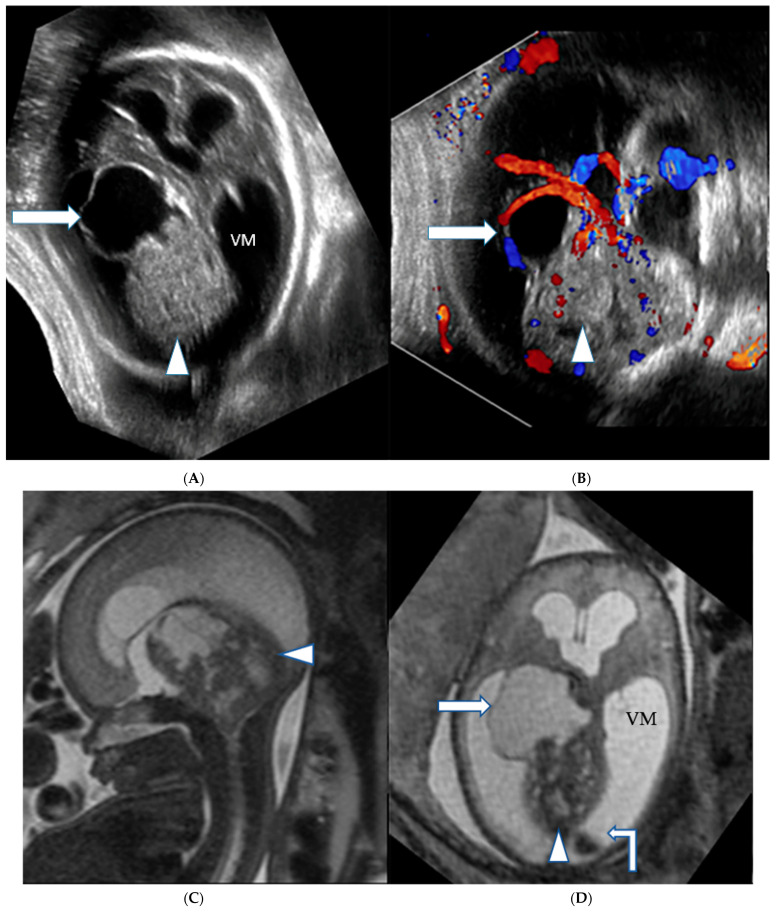
(**A**,**B**): Gray scale and color-coded Doppler sonography US shows a large heterogeneous hyperechogenic solid (arrowhead), partially hypoechogenic cystic (arrow) mass lesion filling out the posterior fossa with compression of the brainstem, cerebellum and vermis. A cystic component herniates through the tentorial incisura into the supratentorial left lateral ventricle. The lesion appears well perfused on Doppler ultrasound. (**C**,**D**): Matching T2-weighted fetal MRI confirms the large posterior fossa mass lesion (arrowhead) with resultant high grade supratentorial ventriculomegaly (VM). A large compared to the cerebrospinal fluid slightly T2-hypointense cyst extends into the left lateral ventricle indicating a higher protein content, possibly hemorrhage within the cyst (arrow). A blood-cerebro-spinal fluid level is noted in the right lateral ventricle (angeled arrow).

**Figure 4 jcm-12-00058-f004:**
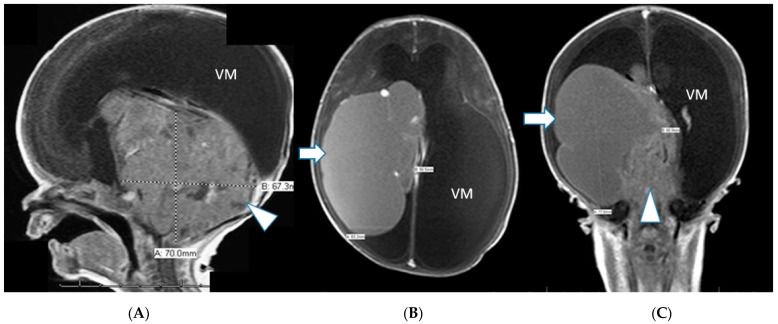
(**A**–**C**): Postnatal sagittal, axial and coronal T1-weighted contrast enhanced MRI showing an interval progression in tumor size in comparison to prenatal images, measuring 7 cm in diameter with near complete compression and infiltration of the cerebellum and brainstem (arrowhead). Large supratentorial, intraventricular protein rich tumor cyst similar to the prenatal imaging (arrow). Ventriculomegaly (VM).

**Figure 5 jcm-12-00058-f005:**
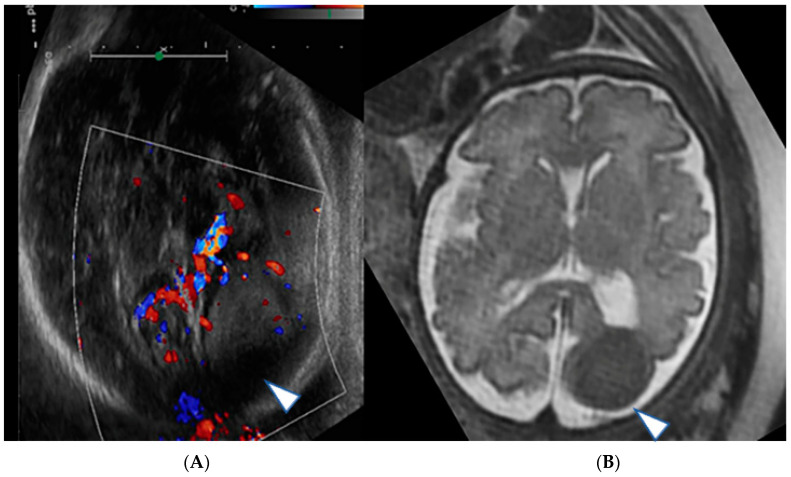
(**A**,**B**): Prenatal color coded axial US image and matching T2-weighted fetal MRI show a solid cortical/subcortical well circumscribed mass lesion within the left occipital lobe (arrowhead).

**Figure 6 jcm-12-00058-f006:**
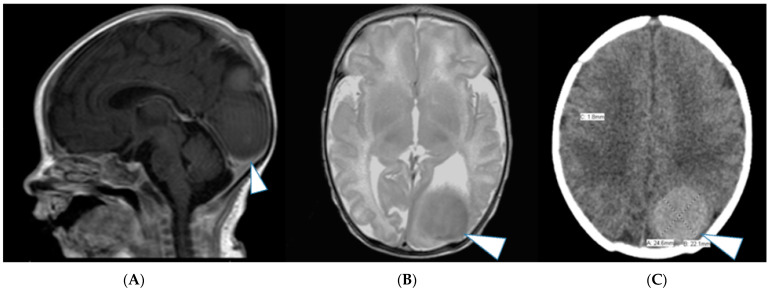
(**A**–**C**): Matching contrast enhanced sagittal contrast enhanced T1-weighted, axial T2-weighted and axial CT images confirm the left occipital lobe highly cellular (T2-hypointense), non-enhancing mass lesion with mild hyperdensity due to the high cellularity compatible with a neuroglial tumor (arrowhead).

**Figure 7 jcm-12-00058-f007:**
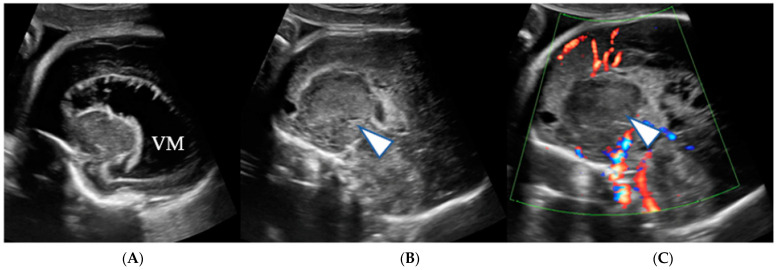
(**A**–**C**) Prenatal gray scale and color-coded Doppler US (**A**–**C**) and axial T2-weighted fetal MRI (**D**), postnatal contrast enhanced T1-weighted and postoperative follow up T2-weighted MRI show a large lobulated solid mass lesion in the left frontal lobe (arrowhead). Post-surgery the tumor has become T2-hyperintense cystic. Ventriculomegaly (VM) is secondary to compression of the foramina Monro (**E**,**F**).

## Data Availability

The analyzed data sets generated during the study are available from the corresponding author on reasonable request.

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
