# Peer review of "Fetal Brain Tumors, a Challenge in Prenatal Diagnosis, Counselling, and Therapy"

_jcm, 2022, doi:10.3390/jcm12010058_

Round 1

Reviewer 1 Report

Congratulations on a well written paper!

What is the main question addressed by the research?

·        The authors have addressed the topic of foetal brain tumours through a series of prenatally diagnosed cases.

Is it relevant and interesting?

·        The paper and the data presented are relevant and interesting since the paper is talking about clinical entities that are rare and difficult to counsel. The data presented can be interesting to all professionals that are involved in the process of counselling parents: foetal medicine specialists, neonatologists, and foetal surgeons….

How original is the topic?

·        The topic is not original since there have been case reports and small series of foetal brain tumours but I have to point out that with the clinical scenario of a foetal brain tumour any new data is welcomed.

What does it add to the subject area compared with other published material?

·        Most importantly the paper states the difficulty of prenatal diagnosis by ultrasound points the clinician to use additional diagnostic modalities like MR and once again shows us how diverse the clinical outcome can be.

Is the paper well written?

·        The paper is well written with a concise Introduction and a detailed presentation of 4 different cases of fetal brain tumours. In the discussion part of the paper, the authors have touched the main point associated with this rare clinical scenario.

Is the text clear and easy to read?

·        Yes

 Are the conclusions consistent with the evidence and arguments presented?

·        The conclusion is logical and derived from the data presented.

 Do they address the main question posed?

·        The authors have addressed the main question posed in the title and have described in an appropriate way.

Author Response

We thank reviewer 1 for his kind comments on our work.

Reviewer 2 Report

This is a well written and illustrated case series of 4 cases of prenatal tumors. I have only one concern with the article: no pathological diagnosis in some of the cases. The authors appropriately acknowledge this limitation in the discussion section of the manuscript.

Author Response

We thank reviewer 2 for the kind evaluation of our article

Reviewer 3 Report

Authors present four fetuses with matching prenatal US and MRI diagnosis of Fetal Brain Tumors presenting at their center between 2011-2019 to familiarize physicians with the various primary brain tumors that can be encountered on prenatal imaging and the respective value of US and MRI to narrow down differential diagnosis. The caae series is very interesting and informative.case Yhis series exemplifies various entities of FBT, the course of pregnancy and their postnatal outcome. Mortality is high and overall survival rate can be as low as 16%, depending on the histopathology, location and growth pattern of the tumor. In this case series, two infants suffered neonatal demise (case 1+2). Autopsy and histopathological confirmation of prenatally suspected tumor was rejected in these cases, limiting diagnosis to pre- and postnatal imaging which is expected.The tile. abstract, index words, introduction, care series description, discussion and conclusion all appear reasonble. References are up to date and relevant, Illustrations are excellent and may be improved by marking the areas of intrest to highlight for ease of spotting the pathology.

Author Response

We thank reviewer 3 for the kind evaluation of our manuscript.

llustrations are excellent and may be improved by marking the areas of intrest to highlight for ease of spotting the pathology.

Response 1: thank you very much for this suggestion. We have highlighted the areas of interest in the illustraations as requested.

Reviewer 4 Report

The topic faced from Authors is really interesting. This case series is limitated in number, but this aspect is justified by the rarity of FBTs.

I think that Authors should better specify the level of competence of their Institution (Is it a referral center for fetal diagnosis? Is it a third level center? ... Probably yes, due to the complexity of the cases reported, but I think it is important to include this information) .

I appreciate the brief schematic description of each case separately with corresponent US and MRI images. The description is clear and fluent.

I believe that case series reports, especially in topic of rare occurence like this, are important to increase knowledges and clinical experiences, and thus help clinicians in future similar management.

Obviously, interdisciplinary management of complex cases is crucial in order to optimize the outcome.

Author Response

We thank reviewer 4 for the kind evaluation of our manuscript.

I think that Authors should better specify the level of competence of their Institution (Is it a referral center for fetal diagnosis? Is it a third level center? ... Probably yes, due to the complexity of the cases reported, but I think it is important to include this information).

Response: We thank reviewer4 for this suggestion and have added the level of our center, "high complexity tertiary referral center"in the manuscript line 64